# Antibacterial Activity and Pharmacokinetic Profile of a Promising Antibacterial Agent: 22-(2-Amino-phenylsulfanyl)-22-Deoxypleuromutilin

**DOI:** 10.3390/molecules25040878

**Published:** 2020-02-17

**Authors:** Xiangyi Zuo, Xi Fang, Zhaosheng Zhang, Zhen Jin, Gaolei Xi, Yahong Liu, Youzhi Tang

**Affiliations:** 1Guangdong Provincial Key Laboratory of Veterinary Pharmaceutics Development and Safety Evaluation, College of Veterinary Medicine, South China Agricultural University, No. 483 Wushan Road, Tianhe District, Guangzhou 510642, China; zuoxiangyi@stu.scau.edu.cn (X.Z.); 15778310073@163.com (X.F.); zhaoshengzhang@scau.edu.cn (Z.Z.); jinzhenhami@scau.edu.cn (Z.J.); gale@scau.edu.cn (Y.L.); 2Technology Center for China Tobacco Henan Industrial Limited Company, No. 8 The Third Street, Economic & Technology Development District, Zhengzhou 450000, China; 3Guangdong Laboratory for Lingnan Modern Agriculture, Guangzhou 510642, China

**Keywords:** pleuromutilin, antibacterial activity, acute toxicity, pharmacokinetic, MRSA

## Abstract

A new pleuromutilin derivative, 22-(2-amino-phenylsulfanyl)-22-deoxypleuromutilin (amphenmulin), has been synthesized and proved excellent in vitro and in vivo efficacy than that of tiamulin against methicillin-resistant *Staphylococcus aureus* (MRSA), suggesting this compound may lead to a promising antibacterial agent to treat MRSA infections. In this study, the effectiveness and safety of amphenmulin were further investigated. Amphenmulin showed excellent antibacterial activity against MRSA (minimal inhibitory concentration = 0.0156~8 µg/mL) and performed time-dependent growth inhibition and a concentration-dependent postantibiotic effect (PAE). Acute oral toxicity test in mice showed that amphenmulin was a practical non-toxic drug and possessed high security as a new drug with the 50% lethal dose (LD_50_) above 5000 mg/kg. The pharmacokinetic properties of amphenmulin were then measured. After intravenous administration, the elimination half-life (T_1/2_), total body clearance (Cl_β_), and area under curve to infinite time (AUC_0→∞_) were 1.92 ± 0.28 h, 0.82 ± 0.09 L/h/kg, and 12.23 ± 1.35 μg·h/mL, respectively. After intraperitoneal administration, the T_1/2_, Cl_β/F_ and AUC_0→∞_ were 2.64 ± 0.72 h, 4.08 ± 1.14 L/h/kg, and 2.52 ± 0.81 μg·h/mL, respectively, while for the oral route were 2.91 ± 0.81 h, 6.31 ± 2.26 L/h/kg, 1.67 ± 0.66 μg·h/mL, respectively. Furthermore, we evaluated the antimicrobial activity of amphenmulin in an experimental model of MRSA wound infection. Amphenmulin enhanced wound closure and promoted the healing of wound, which inhibited MRSA bacterial counts in the wound and decreased serum levels of the pro-inflammatory cytokines TNF-α, IL-6, and MCP-1.

## 1. Introduction

*Staphylococcus aureus* is a major contributor to nosocomial infections worldwide. The pathogen is responsible for a variety of life-threatening diseases, such as sepsis, pneumonia, toxic shock syndrome, and endocarditis [1]. MRSA is *S. aureus* that displays resistance to methicillin and other β-lactam antibiotics, such as oxacillin, nafcillin, and carbapenems. The dissemination of MRSA represents a significant global health issue, which impacts patients in both community and health care settings [2]. The emergence of MRSA infection causes higher costs, longer hospitalization courses, and higher morbidity and mortality [3]. MRSA alone is estimated to cause about 19,000 deaths a year in the United States [4]. The World Health Organization (WHO) had released a list of the drug-resistant bacteria that poses the greatest threat to human health and for which new antibiotics are desperately needed [5]. MRSA is one of the listed bacterium and is divided into the high categories according to the urgency of need for new antibiotics [6]. It is urgent to find new antibiotics that have different antimicrobial mechanisms from existing drugs to avoid cross-resistance.

Pleuromutilin (Figure 1), a naturally tricyclic diterpenoid natural antibiotic, was first isolated from *Pleurotus mutiliz* and *Pleurotus passeckeranius* in 1951 [7]. The pleuromutilin class of antibiotics displayed a distinct mode of action which selectively inhibits bacterial protein synthesis through interaction with the 23S rRNA of the 50S ribosome subunit [8]. The unique mechanism of action made it an attractive target to develop novel antibiotics for the treatment of multidrug-resistant bacterial infections [9].

A large number of semisynthetic pleuromutilin derivatives have been designed and synthesized [10,11] to develop novel pleuromutilin antibiotics with stronger antibacterial activity [12,13], better water solubility [14,15], and higher bioavailability [16,17]. Up to now, four kinds of pleuromutilin antibiotics, including tiamulin, valnemulin, retapamulin, and lefamulin, have been approved by Food and Drug Administration (FDA). Tiamulin and valnemulin were used as therapeutic agents for veterinary clinical use [18,19]. Retapamulin was approved as a topical antibacterial agent to treat skin infection for human use [20]. In 2019, lefamulin (Figure 1), formally BC-3781, was approved to treat the community-acquired bacterial pneumonia (CABP) in adults [21]. As the first intravenous and oral antibiotic with a novel mechanism of action approved by the FDA in nearly two decades, lefamulin attached the attention of pharmaceutical researchers and aroused a research upsurge to the pleuromutilin.

Previous work in our group has made series of novel pleuromutilin derivatives with moderate antibacterial activity against *S. aureus* ATCC 29213 and MRSA [22]. One of these derivatives, 22-(2-amino-phenylsulfanyl)-22-deoxypleuromutilin (amphenmulin, Figure 1), showed excellent in vitro and in vivo efficacy than that of tiamulin against MRSA [22], which motived us to further study its preliminary pharmacological effect. Thus, a series of studies were carried out, including in vitro antibacterial activity, acute oral toxicity and pharmacokinetic studies in mice via three different administration routes. In addition, the therapeutic effect of amphenmulin in an experimental model of MRSA wound infection was also evaluated to investigate its anti-inflammatory efficacy.

## 2. Results

### 2.1. Biological Evaluation

To evaluate the antibacterial activity, the minimal inhibitory concentrations (MICs) of amphenmulin were evaluated against *S. aureus* (ATCC 29213), *E. coli* (ATCC 25922), *S. typhimurium* (ATCC 14028), *E. faecalis* (ATCC 29212), *E. faecium* (ATCC 35667), and five clinical strains of MRSA using tiamulin as positive controls. The results of these studies were summarized in Table 1.

The results of the time-kill curve experiments of amphenmulin were presented in Figure 2. The raw data were shown in Appendix A (see Appendix A). Amphenmulin displayed bacteriostatic activity on MRSA at the concentration of 4× MIC. The antibacterial effect could not be positively correlated with the increase of drug concentration when the drug concentration exceeded 4× MIC, which showed that amphenmulin induced time-dependent growth inhibition rather than dose-dependent.

The PAE of amphenmulin was then investigated to evaluate its in vitro antibacterial pharmacodynamic activity. The results of PAEs of amphenmulin aganist MRSA ATCC 43300 are shown in Figure 3 and Table 2.

### 2.2. Acute Oral Toxicity Study

No mice died after receiving a single-dose of 5000 mg/kg of amphenmulin. A high dose of amphenmulin could cause some abnormal symptoms in mice, such as increase in the heart rate, excitement of some mice and inappetence after administration. All mice recovered in a few days. The LD_50_ of amphenmulin is more than 5000 mg/kg.

### 2.3. Pharmacokinetic Analysis

The pharmacokinetic properties of amphenmulin were measured in mice administrated intravenously, intraperitoneally and orally at a dose of 10 mg/kg. The plasma concentration values at each time point were shown in supplemental data (see Appendix A). The plasma concentration-time curve of amphenmulin via three administration routes were shown in Figure 4, from which the pharmacokinetic parameters could be acquired by non-compartmental analysis (see Table 3).

### 2.4. The Therapeutic Effect in an Experimental Model of MRSA Wound Infection

At one and three days after treatment, the area of wound in mice was measured (see Figure 5). It can be seen that the wound area expanded after the wound infection occurs, and decreased gradually with the wound healing. After treatment for three days, the wound area of the uninfected group was 0.38 ± 0.06 cm^2^, while that of the MRSA infection group, amphenmulin treatment group, and vancomycin treatment group were 0.64 ± 0.08 cm^2^, 0.47 ± 0.09 cm^2^, and 0.44 ± 0.07 cm^2^, respectively.

The wound tissues of mice were taken for bacterial count after treatment for one day and three days (see Figure 6). There were almost no bacteria in the negative group, while in the infected group was about 10^7^ CFU/mL. A significant decrease of the number of bacteria in the amphenmulin treatment group and vancomycin treatment group was observed at one day after treatment, compared with that in the infected group (*p* < 0.05). At three days after treatment, the number of bacteria of the amphenmulin treatment group was 4.47 ± 0.66 log_10_ CFU/mL, and that of the vancomycin treatment group was 4.25 ± 0.57 log_10_ CFU/mL.

It has been reported that valnemulin can inhibit TNF-α and IL-6 released by macrophages induced by lipopolysaccharide, indicating that valnemulin has certain anti-inflammatory effect [23]. For both valnemulin and amphenmulin were pleuromutilin derivatives, we are wondering whether amphenmulin could possess anti-inflammatory effect like valnemulin. Thus, the anti-inflammatory effect of amphenmulin was also investigated in this MRSA wound infection model. The contents of three inflammatory factors, TNF-α, IL-6 and MCP-1 in the serum were determined by ELISA at 24 h after treatment (see Figure 7). MRSA infection led to strong induction of TNF α, IL-6 and MCP-1 in MRSA infected group in comparison to those in uninfected group (*p* < 0.05). At three days after treatment, TNF-α, IL-6, and MCP-1 in the serum of the amphenmulin treatment group were significantly decreased than those of the infection group (*p* < 0.05).

## 3. Discussion

The results of MIC demonstrated that amphenmulin possessed superior in vitro antibacterial activity against MRSA and ATCC 29213 than tiamulin, which is consistent with our previous work [24]. It could also be observed that amphenmulin showed potent activity against all Gram-positive bacteria, except *E. faecalis* (ATCC 29212), and pretty weak activity against the Gram-negative bacteria *E. coli* (ATCC 25922) and *S. typhimurium* (ATCC 14028). Previous study has shown that valnemulin only showed bacteriostatic effect on *Staphylococcus aureus* [25]. According to the time-kill curve, amphenmulin belongs to bacteriostatic agents as well. The PAE of tiamulin at the concentration of 4× MIC after exposure for 1 h and 2 h were 1.90 h and 2.04 h, respectively, which was obtained directly from the previous research [26]. As shown in Table 3, amphenmulin displayed a concentration-independent PAE and performed a comparable or longer PAE than tiamulin against MRSA. It indicated that amphenmulin may possess longer dosing intervals, fewer adverse effects, and lower costs while formulating a daily administration dosage [25].

According to the guiding principle of acute toxicity test (LD50 determination) of veterinary drugs, the LD_50_ of amphenmulin is more than 5000 mg/kg which indicates that amphenmulin is practically non-toxic and safe for clinical use [27].

Inspired by its excellent in vitro antibacterial activity against MRSA and *Staphylococcus aureus*, the pharmacokinetic properties of amphenmulin were examined to determine its potential for further investigation. The results showed that amphenmulin is a fast eliminating drug with the short T_1/2_ and MRT for three administration routes. The C_max_ of amphenmulin after oral administration is significantly lower than that of intraperitoneal injection, which may be due to the obvious first-pass effect of the drug [28]. The T_max_ of intraperitoneal injection was shorter than that of oral administration, indicating that the absorption of intraperitoneal administration was faster than that of oral administration. Area under curve to infinite time (AUC_0→∞_) can describe the amount of drug entering the circulation of the body, which is an important index to evaluate the plasma exposure levels of drug [29]. The limited oral bioavailability of amphenmulin may resulted from its strong hydrophobic nature [30], which was consistent with the poor absorption and low plasma exposure levels of amphenmulin in the organism. Therefore, the structure of amphenmulin needs further optimized to improve its pharmacokinetic profiles. According to our previous work, amphenmulin showed higher in vivo antibacterial activity than that of tiamulin against MRSA 43300 in the mouse infection model [24]. However, the pharmacokinetic properties of amphenmulin were unsatisfactory. The promising in vivo activity might due to its superior MIC, which could achieve potent activity even at a very low drug concentration.

To make a more comprehensive evaluation of its antibacterial effect, we investigated the effect of amphenmulin on the treatment of MRSA infected wounds as an external drug. The results showed that the bacterial burden in the wound of the infected group was stable at 10^7^ CFU/mL, which proved that the local infection was successful. The wounds were treated with amphenmulin using vancomycin as the control group. At three days after treatment, the number of bacteria and the area of wound decreased more significantly in both amphenmulin treatment group and vancomycin group. It indicated that amphenmulin could attenuate the propagation of MRSA in wound and promote wound healing, which is comparable to vancomycin.

Some antibacterial agents not only could regulate immune and inflammatory responses but also possessed antibacterial activity [23]. Valnemulin was proved to have effect on the inflammatory responses for the treatment of LPS-induced acute lung injury (ALI) in mice besides antibacterial activity [31]. Amphenmulin, as a structural analogue of valnemulin, might possess the same effect. Invasive *S. aureus* infection induced the production of localized and systemic pro-inflammatory cytokines, such as TNF-α and IL-6 [32]. The chemoattractant MCP-1 plays an important role in monocytes recruitment to the site of infection, and helps proteins transport the endothelial barrier during wound healing [33]. Therefore, the content of TNF-α, IL-6, and MCP-1 in serum were measured to evaluate the effect of amphenmulin on inflammation in the treatment of an experimental model of MRSA wound infection. The results showed that amphenmulin has the comparable effect as vancomycin in reducing the concentrations of TNF-α, IL-6 and MCP-1 in the serum of mice. Combined with our experimental results, amphenmulin possibly enhanced the wound healing by controlling the inflammatory response of mice [34].

Proper inflammatory response is essential to eradicate infectious microorganisms [35]. However, excessive inflammation could cause damage to the body with a large number of activated immune cells and the excessive release of inflammatory factors [36,37]. At three days after treatment, amphenmulin could significantly decreased TNF-α, IL-6, and MCP-1 in the serum in comparison to those of the infection group, avoiding the adverse effects of excessive inflammatory factors on wound healing. Therefore, amphenmulin, displaying both antibacterial activity and anti-inflammatory effect, has promising application in the treatment of MRSA wound infection.

## 4. Materials and Methods

### 4.1. Chemicals

Amphenmulin (purity: 98%) was synthesized using the existing method [24] with slight modifications in our lab. Tiamulin (purity: 95%) and vancomycin hydrochloride (purity: 95%) were purchased from Guangzhou Xiangbo Bio-Technology Co., Ltd. (Guangzhou, China). All the other reagents and solvents were obtained commercial and were used as supplied.

### 4.2. Microorganisms

Methicillin-resistant *Staphylococcus aureus* (MRSA) ATCC 43300, *Staphylococcus aureus* ATCC 29213, *Escherichia coli* ATCC 25922, *Salmonella typhimurium* ATCC 14028, *Enterococcus faecalis* ATCC 29212, and *Enterococcus faecium* ATCC 35667 were obtained from the China Institute of Veterinary Drug Control (Beijing, China). Five clinical strains of MRSA (N7, N9, N20, N30, and N54) were isolated from diseased pig tissues of South China Agricultural University and Foshan University Animal Hospital. All clinical strains were identified as MRSA by Gram stain, catalase reaction, coagulase test, staphylococcus API identification system test, MecA gene test, and methicillin MIC test.

### 4.3. Animals

Specific pathogen free (SPF) Kunming mice (18–22 g, half male and half female) and Institute of Cancer Research (ICR) mice at SPF grade (23–27 g, six weeks old, female) were purchased from Guangdong Medical Laboratory Animal Center (Guangzhou, China). Female and male mice were kept in different cages with free access to food and water. All mice were acclimatized to the new environment for at least one week before experiment. All experimental procedures were consistent with the Ethical Principles in Animal Research and were approved by the Committee for Ethics in Guangdong Medical Laboratory Animal Center (number: SCXK2013-0002).

### 4.4. Biological Evaluation

#### 4.4.1. Minimum Inhibitory Concentration Testing

The MIC of amphenmulin against MRSA, *Escherichia coli*, *Salmonella typhimurium*, *Enterococcus faecalis*, and *Enterococcus faecium* were determined using tiamulin as controls. MIC values were measured by the method of previous work [38]. *Staphylococcus aureus* ATCC 29213 was used as the quality control bacteria, and the quality control range was referred to the Clinical and Laboratory Standards Institute (CLSI). The bacteria solution was diluted to ~10^6^ CFU/mL with Mueller–Hinton (MH) broth for MRSA, *Escherichia coli* and *Salmonella typhimurium* while Trypticase Soy Broth for *Enterococcus faecalis* and *Enterococcus faecium*. Drugs were dissolved in an aqueous solution of N, N-dimethylformamide (DMF) at a concentration of 2560 µg/mL, and then diluted with broth to prepare working solution (1280 µg/mL). The drug susceptibility testing was performed in 96-well plate. A total of 20 µL of working solution was added into column 1 and well mixed as a 10-fold dilution, following which Serial 2-fold dilutions were prepared to provide concentration ranges of 128–0.0078 µg/mL. Blank broth (no cultures) and bacteria solution (drugs replaced by blank solvent) were set as drug-free controls. Each 96-well plate was cultured at 37 °C for 24 h. The MIC value is the lowest concentration of amphenmulin which inhibits the visible growth of bacteria. The experiment was repeated three times.

#### 4.4.2. Constant Concentration Time-Kill Curves

Time–kill curve experiments were performed in triplicate according previous report [39]. In this experiment, MRSA, in logarithmic phase, was diluted to 10^6^ CFU/mL with MH broth. Various concentrations of amphenmulin were added into the bacteria solution with final concentration of 0.5× MIC, 1× MIC, 2× MIC, 4× MIC, 8× MIC, 16× MIC, and 32× MIC, respectively. The mixtures and control which was free of drug were cultured at 37 °C with shaking. A total of 20 µL aliquots from solutions were serially diluted 10-fold in 0.9% saline, at different time intervals (0, 3, 6, 9, and 24 h). Then the dilutions were plated onto MH agar medium and incubated at 37 °C for 18–24 h. The bacterial colonies were counted, and results were represented as the logarithm of the colony numbers. The time-kill curve was constructed by plotting log CFU/mL versus time.

#### 4.4.3. The Postantibiotic Effect

The PAE was determined with previously methods [40]. MRSA in logarithmic phase was diluted to 10^6^ CFU/mL using MH broth as the inoculum. 0.4 mL amphenmulin reserve solutions, at the initial concentrations of 10× MIC and 40× MIC, were then added to 5.6 mL inoculum, respectively. The group in which the drug was replaced by 0.9% saline was served as the control. All the samples were incubated at 37 °C for 1 and 2 h. At the end of incubation, the drug was removed by 1000-fold dilution with MH broth which was preheated at 37 °C. After well mixing, the broth was further cultured at 37 °C. 100 µL culture solution was extracted at different time intervals (0, 1, 2, 4, 6, and 8 h). Following a 10-fold dilution, colonies were counted after 20 h of incubation. The kinetic curve was established with the logarithm of the colony number as the ordinate and the time as the abscissa. The PAE was presented in hour and calculated by the equation *PAE* = *T_A_* − *T_C_* [40]. *T_A_* was defined as the culture time required for the number of MRSA to up to 1 log CFU/mL after the drug removal, and *T_C_* was the culture time required for the control group.

### 4.5. Acute Toxicity

The acute toxicity test mainly referred to the previous reports with some modifications [41]. Seven male and seven female KM mice in SPF grade were used for this study. Amphenmulin was formulated in DMSO and given in a single- dose at 5000 mg/kg by gavage, while the control group were given DMSO without drug. The animals were monitored for 14 days and their physical appearance, daily food intake, daily water intake, behavior change, and mortality were observed for 14 consecutive days. The median lethal dose (LD_50_) was calculated by modified Karber method on day 15. The experiment was repeated twice.

### 4.6. Pharmacokinetic Studies

Pharmacokinetic parameters of amphenmulin were measured according to previous reports with some modifications [42]. Each group consisted of eight female SPF-ICR mice. Amphenmulin (1 mg/mL) was administered intravenously, intraperitoneally, and orally at a dose of 10 mg/kg, respectively. Formulation of amphenmulin was a filtered solution in 5% DMSO, 5% Tween-80, and 90% normal saline. Serial samples (each 0.2 mL) were collected from the ocular venous plexus and placed in a plastic centrifuge tube. The blood samples were placed at room temperature for 20 min and transferred to a 4 °C refrigerator for 12 h to promote blood coagulation. Then samples were centrifuged at 2000 r/min for 10 min to get serum. Serum samples were frozen at −20°C. 100 µL of samples was accurately absorbed and mixed with 100 µL of acetonitrile in a 1.5 mL centrifuge tube. The sample was mixed vigorously and centrifuged at 12,000 rpm for 10 min. The supernatant was transferred, filtered through a 0.22 μm cellulose membrane filter and used for LC-MS/MS analysis. Equipment and analysis conditions were shown in supplemental data. The pharmacokinetic parameters of amphenmulin were determined from the mean plasma concentration with noncompartmental analysis in WinNonlin 5.2 software (Pharsight, Mountain View, CA, USA). Data were shown as mean ± standard deviation (SD).

### 4.7. The Therapeutic Effect of Amphenmulin in an Experimental Model of MRSA Wound Infection

#### 4.7.1. Mice Model of MRSA Wound Infection

Six weeks SPF-ICR female mice were used for this study. The Experimental model, inducted of inflammatory in mice by skin excision, was established referring to existing methods [34] with some alterations. The mice were anesthetized with the intravenous injection of 1% sodium pentobarbital. After their hair on the back was shaved, the mice were anesthetized and the back skin was perforated with a biopsy perforator at a depth of fascia injury. The inoculum (50 μL), containing 10^9^ CFU/mL of the indicated microorganism (MRSA ATCC 43300), was applied on the wound for three times at 30 min intervals. 100 μL drug (3 mg/mL) was administrated on the wound at 1 h after inoculation for three consecutive days. A total of 100 μL blank matrix was given to the infected control group and the non-infected group in the same way. The wound was wrapped with sterile gauze. During the treatment, all mice were kept individually and free to eat and drink.

#### 4.7.2. Examination of Wound Healing

##### Determination of Wound Size

The wounds were examined on day 1 and days 3 after treatment. The area of wound were measured according to previous reports with some modifications [43]. In each group, six mice were taken out and anesthetized to take photos and measure the size of the wound as the results.

##### Assessment of Wound Infection

At the end of the examination, mice were anesthetized and killed. Tissue samples, reaching to the muscle membrane, were removed from the middle of the wound surface and homogenized in 0.9% saline. Serially diluted aliquots of homogenate were cultured on high-salt mannitol agar plate at 37 °C for 24 h. The log of the colony number was used for statistical analysis.

##### Serum Levels of TNF-α, MCP-1 and IL-6

After administration for 24 h, blood samples were collected from the ocular venous plexus and were prepared to serum samples as previously described. The contents of IL-6, TNF-α, and MCP-1 in serum were detected using ELISA kits according to the instruction. All ELISA kits were purchased from Shanghai Enzyme-linked Biotechnology Co., Ltd. (Shanghai, China).

#### 4.7.3. Statistical Analysis

SPSS16.0 software was used for statistical analysis, and *t*-test and all data are presented as mean ± SD.

## 5. Conclusions

Amphenmulin exhibited superior in vitro antibacterial activity against *S. aureus* and MRSA than tiamulin. Amphenmulin was found to be a bacteriostatic agent, which induced time-dependent growth inhibition and displayed a concentration-dependent PAE. The results of acute oral toxicity test of amphenmulin in mice showed that amphenmulin was a practical non-toxic drug, which indicated that it was safe for clinical use. Pharmacokinetic study of amphenmulin showed that amphenmulin was a fast eliminating drug with unsatisfactory bioavailability. In further, amphenmulin displayed therapeutic effect in an experimental model of MRSA wound infection, such as promoting wound healing, reducing the bacterial load at the infected site, and reducing the contents of TNF-α, IL-6, and MCP-1 cytokines to a certain extent. These results suggested that amphenmulin merits further studies as a novel antibacterial agent against MRSA infections.

## Figures and Tables

**Figure 1 molecules-25-00878-f001:**
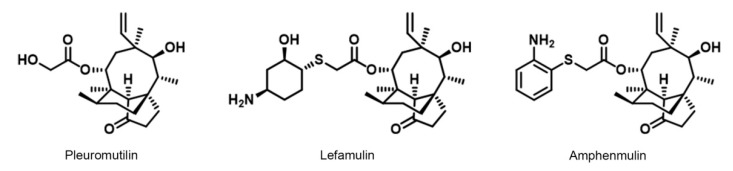
Structure of pleuromutilin, lefamulin, and amphenmulin.

**Figure 2 molecules-25-00878-f002:**
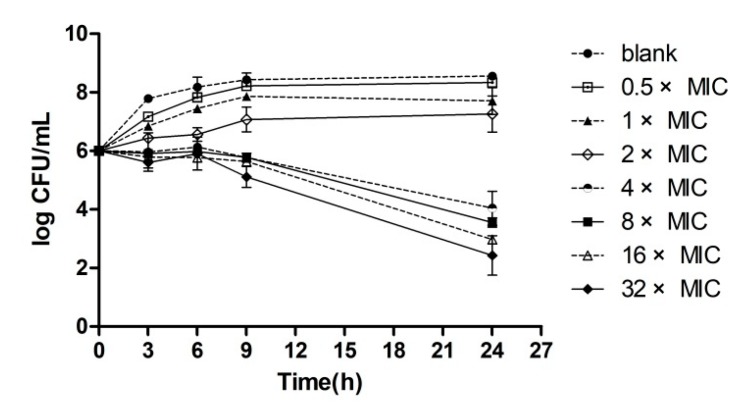
Time-kill curves for MRSA ATCC 43300 with different concentrations of amphenmulin.

**Figure 3 molecules-25-00878-f003:**
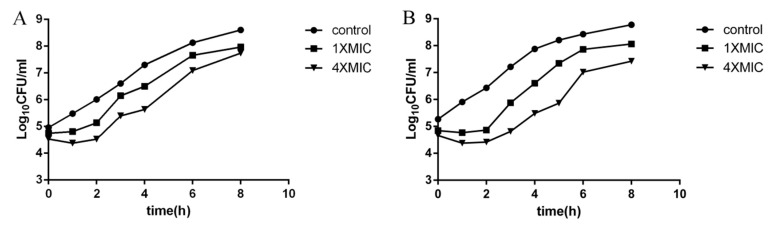
The bacterial growth kinetic curves for MRSA ATCC 43300 exposed to amphenmulin for 1 h (**A**) or 2 h (**B**).

**Figure 4 molecules-25-00878-f004:**
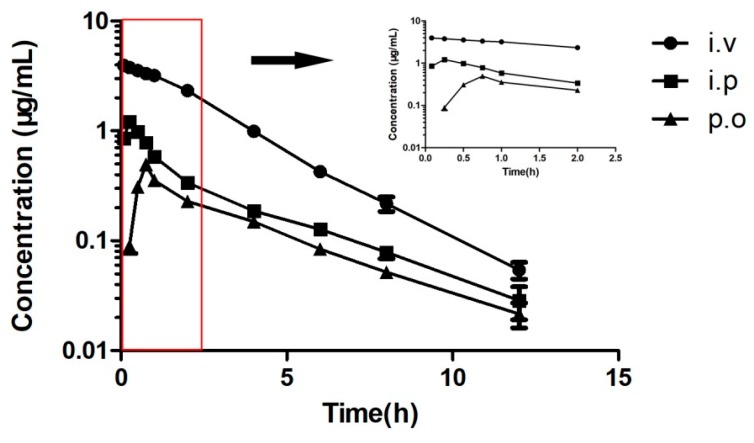
Mean plasma concentration–time curves of amphenmulin in mice after intravenous, intraperitoneal, and oral administration at a dose of 10 mg/kg.

**Figure 5 molecules-25-00878-f005:**
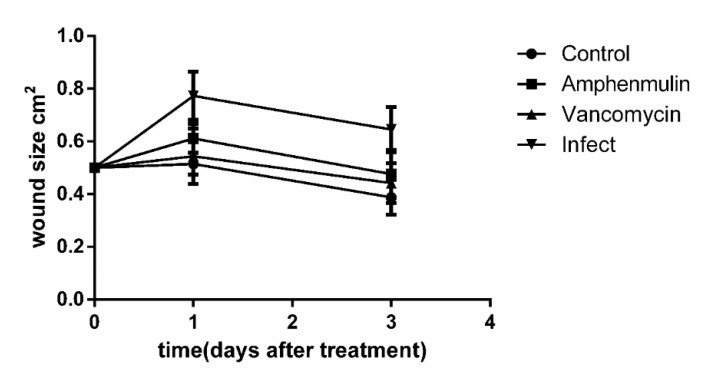
Wound size at one and three days after treatment.

**Figure 6 molecules-25-00878-f006:**
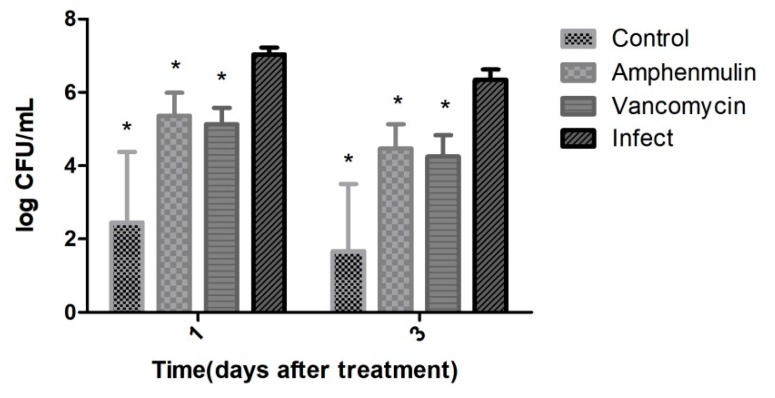
Wound bacterial burden at one and three days after treatment. * 0.01 < *p* < 0.05 vs Infect group.

**Figure 7 molecules-25-00878-f007:**
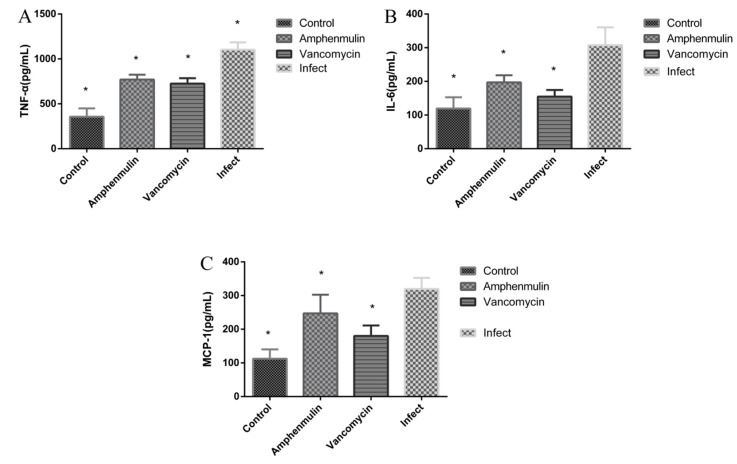
Amphenmulin modulates MRSA mediated-induction of TNF-α (**A**), IL-6 (**B**), and MCP-1 (**C**). * 0.01 < *p* < 0.05 vs Infect group.

**Table 1 molecules-25-00878-t001:** MIC (μg/mL) for *S. aureus* ATCC 29213, *E. coli* ATCC 25922, *S. typhimurium* ATCC 14028, *E. faecalis* ATCC 29212, *E. faecium* ATCC 35667, and five clinical strains of MRSA (N7, N9, N20, N30, and N54).

Compounds	*S. aureus* ATCC 29213	*E. faecalis* ATCC 29212	*E. faecium* ATCC 35667	*E. coli* ATCC 25922	*S. typhimurium* ATCC 14028	N7	N9	N20	N30	N54
Amphenmulin	0.0156	>128	0.125	>128	>128	4	0.0156	4	8	0.0156
Tiamulin	0.5	>128	2	>128	>128	16	0.5	32	32	0.5

**Table 2 molecules-25-00878-t002:** The PAE values of amphenmulin against MRSA ATCC 43300.

Compounds	Concentrations	PAE (h)	
		Exposure for 1 h	Exposure for 2 h
Amphenmulin	1× MIC	0.71	1.35
	4× MIC	1.78	2.83

**Table 3 molecules-25-00878-t003:** Pharmacokinetics parameters of amphenmulin in mice after intravenous, intraperitoneal, and oral administration.

Parameters	Intravenous Administration	Intraperitoneal Administration	Oral Administration
C_max_ (µg/mL)	-	1.34 ± 0.42	0.53 ± 0.18
T_max_ (h)	-	0.32 ± 0.21	1.37 ± 0.74
T_1/2_ (h)	1.92 ± 0.28	2.64 ± 0.72	2.91 ± 0.81
AUC_0→∞_(µg·h/mL)	12.23 ± 1.35	2.52 ± 0.81	1.67 ± 0.66
MRT (h)	2.57 ± 0.19	3.28 ± 0.42	4.01 ± 0.62
CL_β_ (L/h/kg)	0.82 ± 0.09	-	-
CL_β_/F (L/h/kg)	-	4.08 ± 1.14	6.31 ± 2.26
V_z_ (L/kg)	2.17 ± 0.42	-	-
V_z_/F (L/kg)	-	15.92 ± 7.41	28.46 ± 9.01
F%	-	20.71%	13.65%

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
