# Peer review of "Antibacterial Activity and Pharmacokinetic Profile of a Promising Antibacterial Agent: 22-(2-Amino-phenylsulfanyl)-22-Deoxypleuromutilin"

_molecules, 2020, doi:10.3390/molecules25040878_

Round 1
Reviewer 1 Report
In this manuscript was described the biological evaluation of the promising drug candidate (amphenmulin). However, there are some aspects that should be ameliorated:
- In the pharmacokinetic study should be added more detail relative to the volume by weight used in the administration by each one route.
Similarly, more detail about the method used for quantification should be added.
- There are some reasons to use in the studies only female mice?
- It was not clear if there are the formation of metabolites. There are metabolites present during the quantification of the samples?
- More information should be provided regarding the LC-MS/MS method used such as the sample preparation technique used. It was used a previously validated method?
- In the acute toxicity studies seems that the solution of the amphenmulin was prepared only in DMSO but probably could be not the best solvent to consider in these studies in animals. The amphenmulin has low solubility in water?
- In supplementary table 3 there a repetition of the time 0.25 h.
Author Response
Comments and Suggestions for Authors: In this manuscript was described the biological evaluation of the promising drug candidate (amphenmulin). However, there are some aspects that should be ameliorated:
In the pharmacokinetic study should be added more detail relative to the volume by weight used in the administration by each one route.
Similarly, more detail about the method used for quantification should be added.
Response: We are grateful for the valuable comments about this issue. The missing experimental details have been added to the corresponding position in the revised manuscript. According to the given data, it can be calculated that 25g mice need to be injected with 0.25ml amphenicol injection (1mg/mL).
There are some reasons to use in the studies only female mice?
Response: There is no particular reason for using only female mice for pharmacokinetic studies. Compared with male mice, female mice are relatively meek, which is beneficial to complete the collection of blood samples in a short time. There are also a number of studies using female mice for pharmacokinetics. (DOI:10.1016/j.ejps.2019.105158; DOI: 10.1128/AAC.00829-19; DOI: 10.1021/acs.chemrestox.8b00151, et al.)
It was not clear if there are the formation of metabolites. There are metabolites present during the quantification of the samples?
Response: Thank you very much for the opinions which give us new research ideas. For the main purpose of this study is to explore the pharmacological effects of amphenmulin, the metabolites of amphenmulin did not monitored in this study. In the follow-up experiments, we will study further in this area.
More information should be provided regarding the LC-MS/MS method used such as the sample preparation technique used. It was used a previously validated method?
Response: Many thanks your comments. The details of the LC-MS/MS method have been added to the revised supplemental data, and the sample pretreatment method has been added to the corresponding position in the revised manuscript. This is a method that has been verified.
In the acute toxicity studies seems that the solution of the amphenmulin was prepared only in DMSO but probably could be not the best solvent to consider in these studies in animals. The amphenmulin has low solubility in water?
Response: Due to the low toxicity of amphenmulin, the highest dosage needed for the acute toxicity studies is up to 5000mg/kg. Thus, we chose DMSO as the solvent in the acute toxicity studies. Actually, we verified in the pre-experiment that the same dose of DMSO had no significant effect on mice in this study. For the amphenmulin has low solubility in water, it was dissolved in 5% DMSO, 5% Tween-80, and 90% normal saline in other experiment in this manuscript.
In supplementary table 3 there a repetition of the time 0.25 h.
Response: Many thanks your comments. We have deleted the duplicate data in the revised supplementary.

Reviewer 2 Report
In this work, the effectiveness and safety of amphenmulin, a new pleuromutilin derivative was studied. The in vitro antibacterial activity was tested, as well as the in vivo concentration profile in mice after administration via different routes, in order to evaluate the pharmacokinetic parameters. The topic is very interesting and the experimental work well designed and discussed.
In my opinion, only minor changes have to be implemented in the manuscript. They are listed below:
Acronyms in the abstract should be avoided (i.e. PAE, line 23; PK in line 25);
A table of symbols and acronyms used in the manuscript will be of great help to the reader;
Some typos in the manuscript should be corrected (i.e. ‘represent’ – represents line 42, ‘Administration(FDA)’ – Administration (FDA) line 61, ‘and Five clinical’ – and five clinical line 85) Figure 2 is not easily readable.
Please, adjust the graph scale or the figure to distinguish the different data series; In table S3, the time 0.25 h is reported twice, please correct;
Figure 4. The first part of the graph is not easily readable, and this is particularly important for the non-intravenous administration. I suggest a modification of the figure inserting a magnification of the first 2 hours;
Table 3. For my curiosity. Which is the theoretical mean value of the blood concentration immediately after the intravenous injection (i.e. hypothesizing that the drug is instantaneously and uniformly distributed into the blood)?
Line 211,234. Please, explain briefly the modification applied to the previous method in order to make the measurements reproducible.
Author Response
Comments and Suggestions for Authors: In this work, the effectiveness and safety of amphenmulin, a new pleuromutilin derivative was studied. The in vitro antibacterial activity was tested, as well as the in vivo concentration profile in mice after administration via different routes, in order to evaluate the pharmacokinetic parameters. The topic is very interesting and the experimental work well designed and discussed.
In my opinion, only minor changes have to be implemented in the manuscript. They are listed below:
Acronyms in the abstract should be avoided (i.e. PAE, line 23; PK in line 25);
Response: Thanks for this suggestion very much! In accordance with the suggestions, we deleted unnecessary acronyms and explained each acronym that appeared for the first time in the revised manuscript.
A table of symbols and acronyms used in the manuscript will be of great help to the reader;
Response: We are grateful for the valuable comments about this issue. Through statistics, we find that the total number of acronyms and the number of acronyms appear in the full text are not many. We have strictly regulated the use of acronyms in the revised manuscript to make it easier for readers.
Some typos in the manuscript should be corrected (i.e. ‘represent’ – represents line 42, ‘Administration(FDA)’ – Administration (FDA) line 61, ‘and Five clinical’ – and five clinical line 85) Figure 2 is not easily readable.
Response: Many thanks your comments. We re-examined the full text carefully. In addition to the mistakes put forward, we have also corrected other spelling and grammatical errors. In order to make figure 2 more readable, we have adjusted and replaced it in the revised manuscript. However, due to the series drug concentration set in the experiment and a lot of time points, the adjusted figure 2 in the revised manuscript is still not clear enough. Therefore, we put the raw data in the supplemental data to assist readers to read.
Please, adjust the graph scale or the figure to distinguish the different data series; In table S3, the time 0.25 h is reported twice, please correct;
Response: Many thanks your comments. We have made appropriate adjustments to the figures. Duplicate data in the original Table S3 has been deleted in the revised supplementary.
Figure 4. The first part of the graph is not easily readable, and this is particularly important for the non-intravenous administration. I suggest a modification of the figure inserting a magnification of the first 2 hours;
Response: Thanks for this suggestion very much! Figure 4 has been adjusted as recommended in the revised manuscript.
Table 3. For my curiosity. Which is the theoretical mean value of the blood concentration immediately after the intravenous injection (i.e. hypothesizing that the drug is instantaneously and uniformly distributed into the blood)?
Response: It is known that the blood volume of mice accounts for 7% to 8% of their body weight. The concentration of amphenmulin injection we used was 1mg/mL. As the drug was administered at the dose of 10mg/kg, the theoretical mean value of immediate blood concentration of a 25g mice was 0.1mg/ml.
Line 211,234. Please, explain briefly the modification applied to the previous method in order to make the measurements reproducible.
Response: Many thanks your comments. With regard to the modification of the synthetic method of amphenmulin, we only appropriately prolonged the reaction time to obtain a higher yield. Therefore, the experimental steps in this part did not shown in the revised manuscript. For the changes made in the research methods of pharmacokinetic studies, the experimental details have been added to the corresponding position in the revised manuscript and the revised supplementary.
